# Martinize2 and Vermouth provide a unified framework for molecular topology generation

Peter C Kroon[1], Fabian Grünewald[1,2,3]*, Jonathan Barnoud[1,4], Marco van Tilburg[1], Chris Brasnett[1], Paulo CT Souza[1,5,6], Tsjerk A Wassenaar[1], Siewert J Marrink[1]*

[1]Groningen Biomolecular Sciences and Biotechnology Institute, University of Groningen, Groningen, Netherlands; [2]Heidelberg Institute for Theoretical Studies (HITS), Heidelberg, Germany; [3]Interdisciplinary Center for Scientific Computing, Heidelberg University, Heidelberg, Germany; [4]CiTIUS Intelligent Technologies Research Centre, A Coruña, Spain; [5]Laboratoire de Biologie et Modélisation de la Cellule, CNRS, Lyon, France; [6]Centre Blaise Pascal de Simulation et de Modélisation Numérique, Ecole Normale Supérieure de Lyon, Lyon, France

*For correspondence:
Fabian.Gruenewald@h-its.org
(FG);
s.j.marrink@rug.nl (SJM)

Competing interest: The authors declare that no competing interests exist.

## eLife Assessment

The authors present an **important** multi-scale computational platform, which aims to automate the workflow for coarse-grained simulations of biomolecules in the framework of the popular MARTINI model. The capability of the platform has been **convincingly** demonstrated by the application to a large number of proteins as well as macrocycles and polymers. This work will be of interest to both computational biophysicists and chemists.

**Abstract** Ongoing advances in force field and computer hardware development enable the use of molecular dynamics (MD) to simulate increasingly complex systems with the ultimate goal of reaching cellular complexity. At the same time, rational design by high-throughput (HT) simulations is another forefront of MD. In these areas, the Martini coarse-grained force field, especially the latest version (i.e. v3), is being actively explored because it offers an enhanced spatial-temporal resolution. However, the automation tools for preparing simulations with the Martini force field, accompanying the previous version, were not designed for HT simulations or studies of complex cellular systems. Therefore, they become a major limiting factor. To address these shortcomings, we present the open-source *Vermouth* python library. *Vermouth* is designed to become the unified framework for developing programs, which prepare, run, and analyze Martini simulations of complex systems. To demonstrate the power of the *Vermouth* library, the *Martinize2* program is showcased as a generalization of the *martinize* script, originally aimed to set up simulations of proteins. In contrast to the previous version, *Martinize2* automatically handles protonation states in proteins and post-translation modifications, offers more options to fine-tune structural biases such as the elastic network (EN), and can convert non-protein molecules such as ligands. Finally, *Martinize2* is used in two high-complexity benchmarks. The entire I-TASSER protein template database as well as a subset of 200,000 structures from the AlphaFold Protein Structure Database are converted to CG resolution and we illustrate how the checks on input structure quality can safeguard HT applications.

## Introduction

Molecular dynamics (MD) has grown to be a valuable and powerful tool in studying a variety of systems in molecular detail. Advances in force fields and computer hardware have enabled the use of MD in increasingly complex systems, exemplified by recent simulations of, for example realistic cell membranes (*Marrink et al., 2019*; *Yu et al., 2021*), virus particles (*Yu et al., 2021*; *Pezeshkian et al., 2023*), and even complete aerosol droplets (*Dommer et al., 2023*). However, there is a growing interest in studying systems of even greater complexity, culminating in molecularly detailed simulations of whole organelles (*Pezeshkian et al., 2020*; *Autin et al., 2022*) and the set goal of simulating entire cells (*Feig and Sugita, 2019*; *Im et al., 2016*; *Stevens et al., 2023*). Moreover, the growing demand for computer-aided rational design relies on high-throughput simulations with millions of systems simulated in parallel (*Buch et al., 2010*; *Souza et al., 2021b*; *Kutzner et al., 2022*). Currently, the computational demand of MD methods representing all atoms explicitly severely limits the access to the spatial-temporal resolution needed to simulate the aforementioned systems. Coarse-grained (CG) MD methods overcome this challenge by grouping several atoms into one effective interaction site called a bead and thus reduce the number of degrees of freedom that have to be simulated.

Among the most popular CG methods is the Martini force field (*Souza et al., 2021a*; *Marrink et al., 2007*). Within the scope of the Martini force field, about two to five non-hydrogen atoms are grouped into one bead. Nonbonded interactions between beads are defined in a discrete interaction table calibrated to reproduce thermodynamic data, whereas bonded interactions are matched to underlying atomistic reference simulations. Molecule parameters created following this approach are transferable between different systems and chemical contexts (*Souza et al., 2021a*; *Marrink et al., 2007*). This transferability-based approach allows Martini simulations to easily reach the aforementioned complexity scale. However, to really prepare the Martini force field for the high-throughput and whole cell scale simulation era, automated workflows that enable fast and efficient setup of complex systems are of fundamental importance.

The Martini community has a long-standing history of easy-to-use and freely accessible scripts and programs, which helped researchers to set up, run, analyze, and backmap simulations. A non-exhaustive overview can be found in our recent review of the 20-year history of Martini (*Marrink et al., 2023*). However, the codes and scripts developed to date share no common framework or backend even though they share many common operations such as resolution transformation or mapping of coordinates. In addition, input files that define molecule parameters or fragments thereof are not transferable between the tools, with each one of them often defining their own input file formats. We consider that unifying these operations as well as input streams into a single framework will speed up program development and also the robustness of code design to bugs. In addition, it will allow the implementation of modern software techniques such as code review, continuous integration (CI) testing, and version control, which generally improve code quality and resilience (*Abraham et al., 2018*). It is also simpler to adopt a single framework to new developments such as the recently proposed CGsmiles (*Grünewald et al., 2025*) line notation. CGsmiles strings can describe molecules at multiple different resolutions and the interconversion between these resolutions. Thus, they offer a more robust way for sharing, storing, and applying resolution transformations compared to previous data formats.

Designing and coding a unified framework to support general Martini software development is a massive undertaking with many facets as the original scripts and programs deal with different stages of MD simulations. To start the development, we focused the design of the framework on topology generation. A topology lies at the heart of each simulation and defines the starting coordinates as well as input parameters for the simulation. For example, to run protein simulations within Martini, a script called *martinize* (*de Jong et al., 2013*) takes atomistic protein coordinates, maps them to the CG resolution, and generates the protein molecule definitions from building blocks. This workflow is quite classic and underlies many scripts and programs for topology generation both at the CG and at the all-atom (AA) level (Appendix 2) (*de Jong et al., 2013*; *Abraham et al., 2015*; *Páll et al., 2015*; *Case et al., 2005*; *Brooks et al., 2009*; *Phillips et al., 2005*; *Machado and Pantano, 2016*; *Danne et al., 2017*; *Girard et al., 2019*; *Jo et al., 2017*; *Qi et al., 2015*; *Malde et al., 2011*; *Canzar et al., 2013*; *Jorgensen and Tirado-Rives, 2005*; *Dodda et al., 2017b*; *Dodda et al., 2017a*; *Vanommeslaeghe and MacKerell, 2012*; *Uusitalo et al., 2015*; *Jo et al., 2008*; *Uusitalo et al., 2017*). With the latest release of version 3 of Martini, proteins have been thoroughly reparametrized (*Souza et al., 2021a*).

The new capabilities of Martini 3 proteins are exemplified by their use of high-throughput drug binding assays (*Souza et al., 2021b*; *Souza et al., 2020*), which are an essential step in computer-aided drug design (CADD). Part of the improved protein properties comes from the redefined Martini interaction table. However, another part of the improvement is the result of protein-specific methods such as the use of structure-biased dihedrals (*Herzog et al., 2016*) (often referred to as side-chain corrections), specific ENs (*Periole et al., 2009*), or integration of Gō-like models (*Poma et al., 2017*; *Korshunova et al., 2024*; *Pedersen et al., 2024*). All these features are additional specific biasing steps applied after the generation of the original topology file for the protein and are not part of the capabilities of the previous *martinize* script. Hence, we chose to co-develop a unified framework for topology generation together with a new *martinize* version, *Martinize2*.

In this paper, we present the VERsatile MOdular Universal Transformation Helper (*Vermouth*) library, a general Python framework aiding in the design of programs that can create topologies for complex systems at AA, united-atom (UA), and CG resolution. On top of *Vermouth,* we built the *Martinize2* program, as the successor of the *martinize* script (*de Jong et al., 2013*; *Uusitalo et al., 2015*). The goal of *Martinize2* is to encompass all functionality required to generate Martini protein parameters (supporting the older versions Martini 2 *de Jong et al., 2013*; *Periole et al., 2009*; *Monticelli et al., 2008* as well as the latest Martini 3) and be compatible with high-throughput workflows as needed in CADD approaches based on Martini. In addition, both *Vermouth* and *Martinize2* are designed to have sufficient flexibility and robustness to ready Martini for the era of high-throughput high-complexity simulations.

Finally, we note that much of the progress of Martini has resulted from an active community of researchers contributing scripts, programs, and parameters. However, as is the case for most research software in the field, they often fail to adhere to the principles of FAIR: findability, accessibility, interoperability, and reusability. (*Chue Hong et al., 2022*; *Wilkinson et al., 2016*; *Alibay et al., 2022*) The FAIR principles *Wilkinson et al., 2016*, originally designed to improve data management and reproducibility in science, have recently been extended to research software in a more general sense. This extension is aimed at fostering more sustainable software development in science (*Chue Hong et al., 2022*). To meet these standards, the software tools we present here are distributed under the permissive open-source Apache 2.0 license on GitHub and are developed using contemporary software development practices, such as continuous integration testing. To make adoption as easy as possible, they have few dependencies, are distributed through the Python Package Index, and can be installed using the Python package manager *pip*. Other researchers are encouraged and welcome to contribute parameters and code as outlined in our contribution workflow.

## Results

In this section, we first outline the design and API of the *Vermouth* library. Then we discuss how the *Vermouth* library is used to construct a pipeline for generating protein input parameters for the Martini force field. This pipeline constitutes the new *Martinize2* program. Finally, we present some benchmarks and selected test cases to demonstrate the capabilities of *Martinize2* and assess its fitness for generating complex system topologies and high-throughput workflows, surpassing the capabilities of the previous *martinize* script.

### The *Vermouth* library

In the literature, many scripts and programs have been described that can create topologies for linear molecules and some specific software exists that also handles branched molecules such as carbohydrates (*Danne et al., 2017*), or dendritic polymers (*Girard et al., 2019*). However, to the best of our knowledge, there is at present not a general program that can create topologies from atomistic structures for any type of system, and at any resolution, presenting an extendable and stable API. Based on the existing software, we can, however, define a number of required and desirable features for such a general program and library to have: (1) it must be force field and resolution agnostic; (2) it must be MD engine agnostic; (3) it must use data files that can be checked, made, and modified by users, and (4) it must be able to process any type of molecule or polymer, be it linear, cyclic, branched, or dendrimeric, and mixtures thereof.

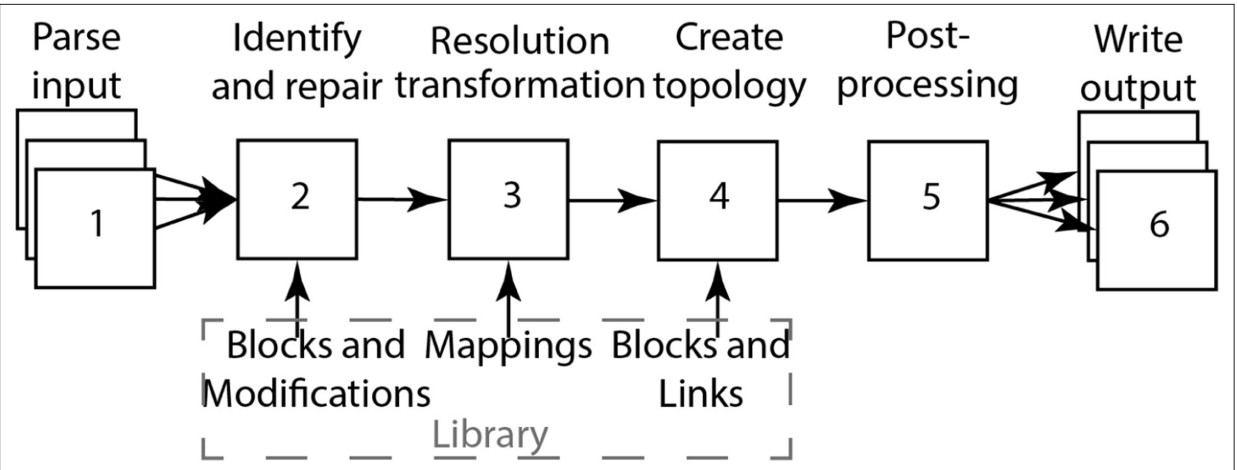

**Figure 1.** Fundamental stages in topology generation from atomistic structures. First, the provided input is parsed (step 1). Second, for every parsed residue, its atoms are identified and, if needed, atom names are corrected and missing atoms are added (step 2). Third, mappings are taken from the library and a resolution transformation to the required output resolution is performed (step 3). Fourth, intra-residue interactions are added from blocks taken from the library, and inter-residue interactions are added from links taken from the library (step 4). Fifth, optionally, post-processing is performed to add, for example an EN (step 5). Finally, the produced topology is written to output files (step 6).

To start designing a library that can fulfill the above requirements, we note that most workflows used for topology generation can be decomposed into six fundamental stages (*Figure 1*): First, reading input data, usually an atomistic coordinate file (e.g. from the protein data bank); second, identifying the parsed atoms, to find how they correspond to the atoms in the data files describing the building blocks; third, optionally a resolution transformation step; fourth, the generation of the actual topology and assigning particle types and bonded interactions; fifth, any type of post-processing; and finally, sixth, writing the required output files. Even though these stages are generally shared for topology generation pipelines, they also apply to other workflows commonly encountered in Martini programs. Especially, stages 1, 3, 5, and 6 can be found in almost all Martini programs, which generate simulation input files in the broader sense (*de Jong et al., 2013*; *Grünewald et al., 2022*; *Empereur-Mot et al., 2020*; *Wassenaar et al., 2014*). Separating these stages, therefore, helps to define an API with data structures and independent processes, which optimally support such workflows. In addition, the clear distinction in stages helps to externalize any data files, which can be edited by the user or force field developers. *Vermouth* is built on the idea and definition of processors, which are tasks arranged in a pipeline. This design was inspired by the ubiquitous workflow managers available in the field (*Marx, 2020*). We formalize the idea of processors by introducing an abstract base class the *Processor*. New pipeline stages can be created as subclasses of this base class. All *Processors* operate on the central data structure class *System*, which contains any number of *Molecule* data structures (see *Figure 2*). A *Molecule* is defined as the graph of a molecule or assembly of molecules, which are connected by bonded interactions. The nodes of a *Molecule* usually correspond to atoms or CG beads but can be any form of particle as defined by the force field.

Nodes can have attributes that describe additional information such as a residue name or charge. However, only the atom name, residue name, and residue number are required as attributes. In addition, the edges of the *Molecule* follow the connectivity as defined by bonds, angles, or other bonded interactions. For example, two protein chains connected by a disulfide bridge would be considered a single *Molecule*. In contrast, a cofactor, which is only interacting via non-bonded interactions, would be a separate *Molecule*. Operations on *Molecules* usually add or remove bonded interactions or change node attributes. For convenience, *Processors* can also operate on a collection of molecules, which are defined by the *System* class (see *Figure 2*). A list of all available processors is given in the documentation.

*Processors* operate on *Molecules*. However, often additional data is required to perform the pipeline as defined by the *Processor*. The additional data can be provided in the form of one of the four other main data structures (*Blocks, Links, Modifications, Mappings*) or arguments of the processors that can be set in a script or via the command line interface. These four other data structures contain

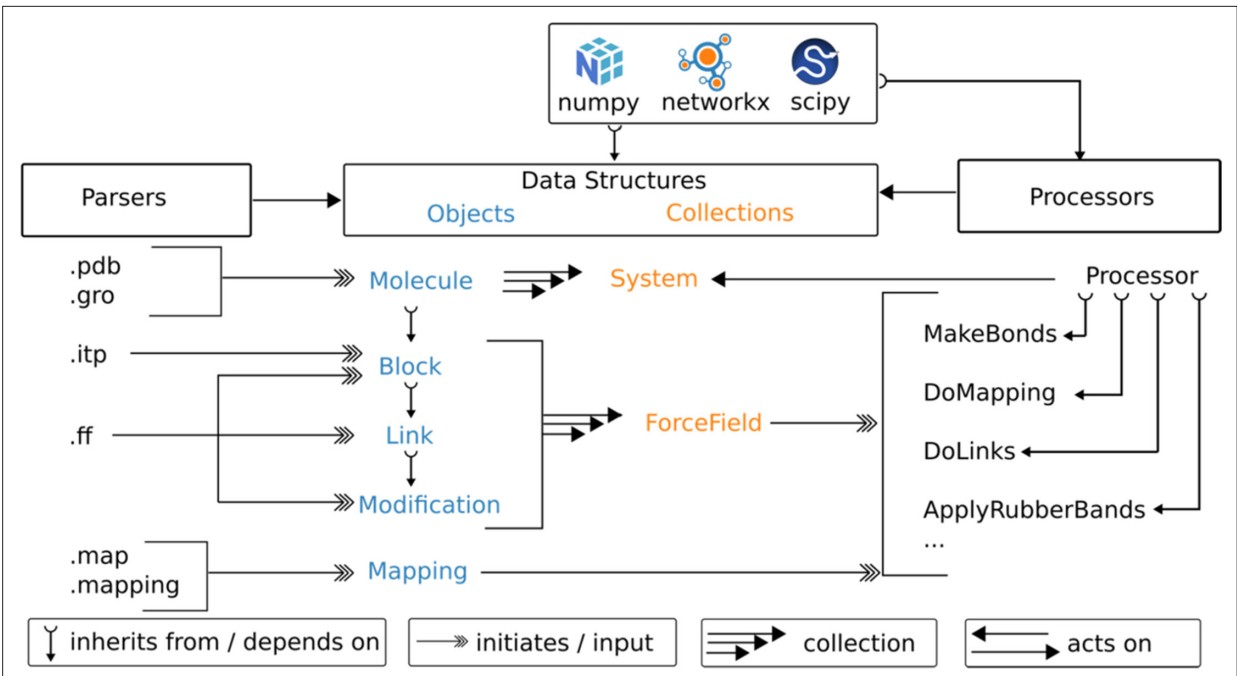

**Figure 2.** Organization of the Vermouth library. The vermouth library defines 5 types of data structures (blue) to store molecular information and force field information. For convenience, it also defines two collection classes (orange) composed of several data-structure instances. Data structures are initiated or get input from parsers, which read 6 types of data files (see *Appendix 1—table 1* for more details on file types). The central data structure(s) are **Molecule** and **System**. These are changed, updated, or transformed by so-called **Processor** classes, which take force field data as input. Parsers, data structures, and **Processors** only depend on three Python libraries as shown. At the moment, vermouth also exposes four types of writers (not shown here) to go along with the parsers (see *Appendix 1—table 2*).

all molecular level information required to fully define and/or modify a topology for any type of MD code (e.g. atom types, bonded interactions, and positions) as well as enable transformations between topologies. For example, a *Mapping* consists of two molecular fragments at different resolutions and a correspondence between their particles. In contrast, *Blocks*, *Links*, and *Modifications* are graphs that describe these molecular fragments, the links between those, and possible changes to fragments, respectively. They are all subclasses of a *Molecule* and an extension of the graph class from the networkx library (*Hagberg et al., 2008*) (see *Figure 2*).

As shown in *Figure 2*, to make the data structures that are force field specific (*Blocks*, *Links*, *Modifications*) easier to use, *Vermouth* offers a second collection class called a *ForceField*. Every molecule must have a *ForceField* associated with it. Additional information on the data structures is given in the documentation.

Finally, the *Vermouth* library also contains a number of parsers that return instances of the data structures from common input file formats. For example, the in-house ff format defines *Blocks*, *Links*, and *Modifications*, while the backwards style mapping format can be read to return an instance of the *Mapping* class. *Appendix 1—table 1* summarizes all input parsers as well as the format and data structure they return. We note that *Vermouth* is also able to read content associated with the '[molecules]' directive of the GROMACS topology file, which is colloquially referred to as included topology file (itp). This allows users to directly manipulate GROMACS molecule files within *Vermouth*. We note that as neither parsing nor the *Molecule* itself depends on GROMACS code, the library can easily be extended to other MD engines.

## Martinize2

*Martinize2* is a pipeline constructed of *Vermouth Processors* with a command line interface (CLI), with the purpose of transforming atomistic structure data to a CG Martini topology including both coordinates and simulation parameters. *Martinize2* is the successor of the *martinize* script, which was used for generating input parameters for Martini 2 proteins, DNA, or RNA. However, different branches

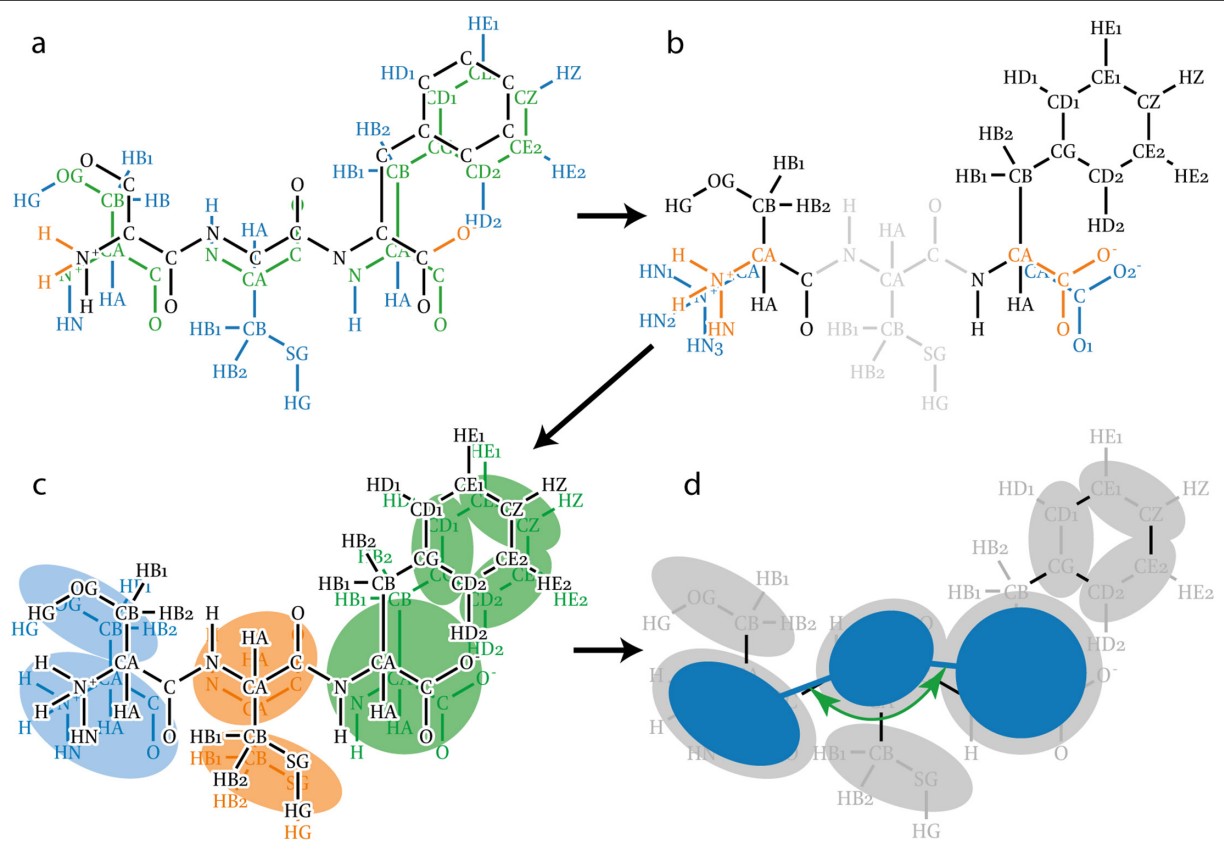

**Figure 3.** Illustration of atom recognition, mapping, and linking in topology generation. (**a**) To identify all atoms in the input molecule (black and orange) every residue in the molecule is overlaid with its canonical reference (blue and green). Atoms are recognized when they overlap with atoms in the reference (green). Atoms not present in the molecule are also identified (blue) and will be added later. Finally, atoms in the molecule not described by the canonical references are also labeled (orange) so that they may be identified later. (**b**) Identifying the terminal atoms that are not part of the canonical residues. The modification templates are depicted in blue and the atoms they match in orange. The cysteine does not participate since it does not carry any unexpected atoms and is depicted in gray for clarity. (**c**) Mappings (blue, orange, and green) describe a molecular fragment at two different resolutions and a correspondence between their particles. The correspondence is depicted approximately here. The mappings are applied to the molecule (black). (**d**) Example of applying a Link. The link depicted (dark blue) adds an angle potential over CG backbone beads.

had to be used for proteins and DNA martinize.py (*de Jong et al., 2013*), martinize-dna.py (*Uusitalo et al., 2015*) or RNA (*Uusitalo et al., 2017*). In contrast, *Martinize2* is designed to generate topologies for the Martini force field for proteins, DNA, and in principle, any other arbitrarily complex molecule.

*Martinize2* consists of different *Processors* which fulfill the basic stages of topology generation as shown in *Figure 1*. We note that the design of *Martinize2* is general and applies to arbitrarily complex polymers consisting of arbitrary monomeric repeat units (MRUs). However, to increase the readability of the following sections, the layout of the program is described in terms of residues in proteins.

The *Martinize2* pipeline starts by reading an atomistic structure, which describes a single molecule (e.g. protein) or assembly of any size. Subsequently, bonds between the atoms are inferred either by distance calculation, atom names within residues, or using CONECT records of the PDB file. All atoms that are connected by bonds form a *Molecule*. Thus, *Martinize2* creates a *System* of *Molecules* at the atomistic resolution at the end of the input reading stage. In stage 2, *Identify and Repair*, each residue of each molecule is compared against its canonical definition. Canonical definitions are selected by residue name from the library files. This comparison identifies missing or additional atoms on a residue and fixes all atom names (*Figure 3a*). To efficiently do these comparison operations, *Martinize2* relies on a number of algorithms coming from graph theory (e.g. subgraph isomorphism), which reduces the dependence on accurate atom names, since these occasionally differ based on the source of the input structure. Which algorithms are used in the code is described in more detail in Appendix 3 and Appendix 4. Once the residues have their canonical atom names, *Martinize2* checks if the missing or

additional atoms are described by any of the modification files (*Figure 3b*). Modifications describe changes in residues from their canonical form, for example different protonation states, termini, or post-translational modifications (PTMs), and the effect these have on the topology.

After completing the repair stage, everything is in place to perform the mapping to CG resolution. The mapping descriptions are read from the mapping input files in the library and tie together residue definitions at the AA and CG level and the correspondence between them (*Figure 3c*). Mapping to CG level in *Martinize2* is done with a multistep subgraph isomorphism procedure, which is general enough to cover edge cases such as when mappings span multiple atomistic residues. A detailed description is provided in Appendix 4. The mapping Processor provides a *System* of *Molecules* at the CG level. These molecules already define all bonded interactions within the residues as well as the coordinates of the CG system. To generate the interactions that link the residues, a simple graph matching with library link definitions is done in the create-topology stage (*Figure 3d*). Finally, after that, we end up with the full CG topology, which is ready for post-processing steps. Post-processing summarizes all biases and modifications that have to be done on the CG molecule and its CG coordinates. For example, an EN is needed to keep the tertiary structure of the protein and is applied in the post-processing stage. Finally, *Martinize2* writes out the CG coordinates and the CG topology file that are production-ready.

## Custom protonation states and PTMs

Of the 20 common amino acids, there are four (GLU, ASP, LYS, HIS), which can readily change their protonation state as a function of pH or environment. Whereas commonly those amino acids are still considered to be in their pH 7 protonation state, it is more appropriate to determine their local

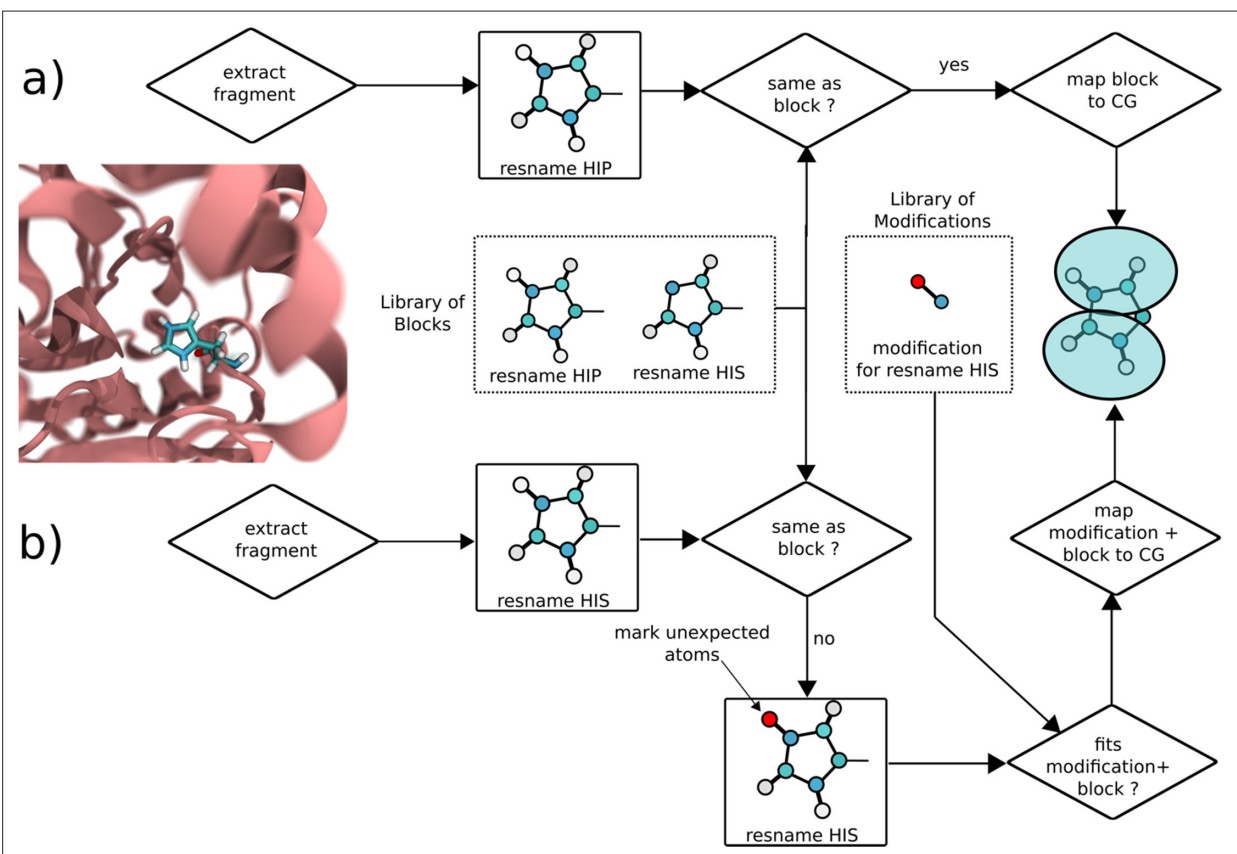

**Figure 4.** Workflows for identifying protonation states or PTMs exemplified on protonated histidine. In route (**a**), the residue name of the protonated histidine extracted from the atomistic coordinates matches the residue name in the library and matches the fragment. Hence, the protonation state is correctly picked up. In route (**b**), the residue name matches that of neutral Histidine in the library. A mismatch of the fragments is recognized, and the extra hydrogen is labeled. Subsequently, by matching the extra hydrogen to a modification of the Histidine block, the protonated Histidine is recognized as neutral Histidine plus protonation modification, and the correct parameters for protonated Histidine are generated.

pKa from, for example continuum electrostatics (*Bashford and Karplus, 1990*). Subsequently, the appropriate charge of the amino acid can be determined from that pKa and set for the simulation. Even though recently more advanced methods became available for dynamically treating protonation states (*Huang et al., 2016*; *Donnini et al., 2011*; *Bennett et al., 2013*) – also at the Martini level (*Grünewald et al., 2020*; *Aho et al., 2022*) – the fixed charge approach is still the most common and for Martini most computationally efficient. However, the previous *martinize* version lacked the functionality to treat protonation states for all amino acids. Only histidine protonation states could be set interactively but only for two of three possible protonation states.

Other protonation states as defined by the atomistic structure coordinates or residue names were ignored without warning. In addition, the interactive setting of protonation states becomes very cumbersome for large protein complexes.

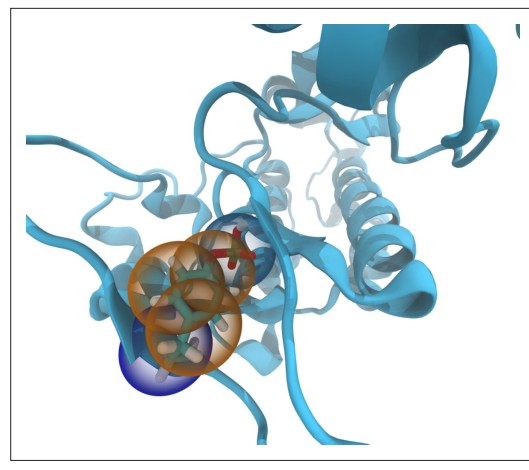

**Figure 5.** Example of automated identification of PTMs. CG Martini model of phosphorylated Tyrosine found in the EGFR kinase activation loop. The mapped structure of the phosphorylated residue is shown as beads overlying the atomistic structure.

To overcome this problem and make protonation state handling easier and more robust, we utilize a dual strategy in *Martinize2* to identify and correctly set the protonation states (see *Figure 4*). In route (a) the user provides atomistic structure coordinates with AA residue names including those of non-default protonation states corresponding to the naming conventions used in CHARMM (*Huang et al., 2017*) or AMBER (*Lindorff-Larsen et al., 2010*). Protonation states can be obtained from online servers such as H++ (*Anandakrishnan et al., 2012*) or propKa (*Olsson et al., 2011*), for example. If the residue names are correctly given, they can be matched against the parameters in the library and the CG residue obtains the correct protonation state. In the alternative route (b), the residue name is simply that of the default pH 7 amino acid; however, the structure file contains an additional hydrogen. In the repair and identify step, the chemical graph of the amino acid is compared to the building blocks in the library, and any unexpected atoms are flagged. For example, in the case of protonated histidine, the additional hydrogen is labeled (see *Figure 4*). Subsequently, *Martinize2* checks if there are any modifications that would match the complete input graph if added to the original block. In the Histidine example, the modification contains the additional hydrogen, which together with the original Histidine block makes up a protonated Histidine. The modification also changes the mapping such that the correct protonation state is set at the CG level. This route is more appropriate for example when processing crystal structure files, which are not necessarily named according to any force field convention. We have tested this feature on two protein structures taken from the PDB (1MJ5, 3LZT) and processed as described in the Methods section. In 1MJ5, there are six Histidine residues, of which one is predicted to be charged at pH 7. The others are neutral. However, they are divided between the ε-tautomer (3 residues) and the δ-tautomer (2 residues). Martini 3 parameters are different for the two tautomers in contrast to Martini 2, which is accordingly recognized by *Martinize2*. In addition, for Lysozyme, we have considered residue GLU35 protonated, which would be appropriate at pH 6 or lower. For both examples, the appropriate protonation states and tautomers are generated at the CG level.

The same procedure used for setting protonation states also applies to identifying any other common PTMs. Using this procedure, lipidation, phosphorylation, amination, or acetylation can be taken into account automatically. To demonstrate that *Martinize2* can handle PTMs, we have implemented dummy parameters for testing of Tyrosine phosphorylations in the Martini 2 force field and generated a Martini topology for the EGFR kinase as an example (PDB 2GS2). Residue TYR845 (see *Figure 5*), which is located in the activation loop of the EGFR kinase, is phosphorylated when the kinase is activated (*Bennett et al., 2013). Martinize2* was able to convert the structure in one go to Martini 2 resolution. We note that at the time of writing, the Martini 3 force field is lacking parameters

for these PTMs, and they are therefore not implemented in *Martinize2* yet. In this case, a warning is issued by the program.

## Expanding the options of elastic network fine-tuning

Due to the limitations in most CG protein models (e.g. lack of explicit hydrogen bonding directionality), the tertiary structure has to be enforced with a structural bias called EN (*Kmiecik et al., 2016*). An EN for Martini proteins consists of weak harmonic bonds between backbone beads of residues (within a chosen cut-off distance) and is generated after the resolution transformation as a postprocessing step (*Periole et al., 2009*; *Monticelli et al., 2008*). *Martinize* offered only two types of EN options, the regular model and the Elnedyn (*Periole et al., 2009*) approach, both of which are also implemented in *Martinize2*. However, as the EN fixes the tertiary structure, changes in the structure upon, for example ligand binding are not captured. To improve protein models in this aspect, recently, Gō-like models have been applied to Martini (*Poma et al., 2017*). In a Gō-like model, the harmonic bonds are substituted by custom Lennard-Jones interactions that can dissociate, thereby allowing for some tertiary structure changes. Within the scope of Martini, a workflow is available to replace the EN with a Gō model that is generated from a provided contact map.

Even though Gō models offer better flexibility, they are currently limited to single monomeric protein units and require some fine-tuning to get the optimal performance (*Poma et al., 2017*). Especially for high-throughput workflows, the EN approach is therefore the preferred option. To further improve upon the ENs generated by the old *martinize*, *Martinize2* offers several options to fine-tune the EN and get better behavior within the constraints of the EN approach. Besides the cut-off and force constant, *Martinize2* now implements a residue minimum distance (RMD). The RMD is defined as a graph distance and dictates how far residues need to be apart in order to participate in elastic

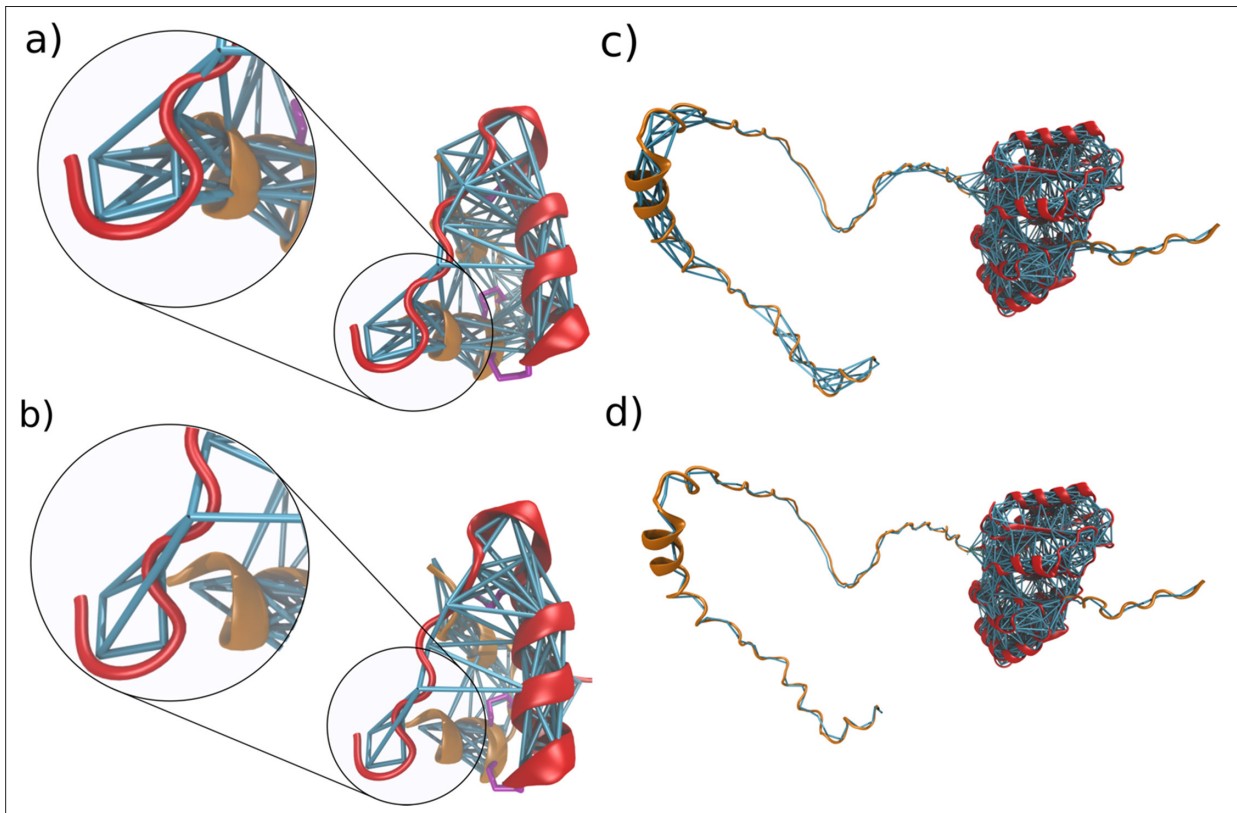

**Figure 6.** Fine-tuning options for the elastic network. (**a**) ENs and backbone bonds within the human insulin dimer when generated with the molecule or all-option. The dimer consists of two chains colored in red and orange, which are connected by two disulfide bridges shown in purple. EN bonds are generated between the two chains and within the chains. (**b**) EN and backbone bonds within the insulin dimer when generated with the chain option. In this case, no elastic bonds are generated between the two chains. They are only connected by the disulfide bridge and non-bonded interactions. (**c**) EN within the FtsZ protein, when generated for both the intrinsically disordered tail domains (orange) and structural domain (red) (**d**) EN within the FtsZ protein when the EN is only generated within the structural domain by defining the EN unit as going from residues 12–320.

bonds. Defining the RMD as a graph distance means that no bonds are generated between residues that are, for example bound by a disulfide bridge. It thus presents a more rigorous implementation than in the previous version. Usually, the residue minimum distance is 3 in order to avoid the EN competing with the bonds, angles, and dihedrals between the backbone beads.

We note that this is part of the Martini protein model and should not be changed. Additionally, *Martinize2* allows you to select which beads to generate the EN between. This option is needed for Martini 2 DNA (*Uusitalo et al., 2015*), for example. Martini 2 DNA offers a stiff EN version, where also sidechain beads are included. Furthermore, *Martinize2* allows defining where in the protein to apply the EN. This is done with the EN unit option. The EN unit can be a molecule, chain, all, or ranges of residue indices. The most trivial option is *all,* in which case an EN is applied between all protein molecules in the system. The option molecule and chain yield the same network, if distinct molecules are also distinct chains. However, when two chains are connected by a disulfide bridge, for example, they would be one *Molecule* in the *Martinize2* sense. On the other hand, if the interface is not very well defined or more flexible, biasing the two chains separately could improve the EN. In that case, the *chain* option can be used. This use case is shown for the human insulin dimer in *Figure 6a and b*. The human insulin dimer consists of two chains, which are connected by two disulfide bridges. If the molecule or all option is used, an EN is generated within the chains and between the chains (*Figure 6b*). However, to avoid generating the EN between the two chains, the chain option can be supplied, in which case the EN is only generated within chains. As the zoom-in on the tail part shows, there are no more bonds between the two chains in *Figure 6b*, whereas there are in *Figure 6a*.

Furthermore, *Martinize2* allows the definition of regions of residue IDs where an EN should be generated. This feature gives maximum flexibility and allows biasing structural regions of proteins, whereas an EN in intrinsically disordered regions (IDRs) can be avoided. For example, *Figure 6c and d* show the FtsZ protein of *Escherichia coli* as predicted by AlphaFold (*Varadi et al., 2022*; *Jumper et al., 2021*). FtsZ possesses a structural unit and two disordered tail domains. With the region option, *Martinize2* allows the generation of an EN only for the structural domain. Within the old *martinize* superfluous bonds needed to be removed manually.

Finally, we note that *Martinize2* is now implemented in the Martini Data Base (MAD), which offers a further utility to remove certain elastic bonds selectively (*Hilpert et al., 2023*). However, ENs can only be applied within protein molecules at the moment.

## Beyond proteins; incorporating other molecules in Martinize2

Legacy *martinize* is only applicable to one category of molecule (i.e. proteins or DNA), which is one of its biggest drawbacks even for setting up simple protein simulations. *Martinize2* allows the inclusion of new classes of molecules without adjusting the codebase. For instance, proteins frequently have other molecular units associated such as ligands, cofactors, metal ions, or lipids. The general workflow of *Martinize2* allows us to convert these systems in one go, provided that the library files are present. Having a single step for topology generation greatly facilitates high-throughput workflows such as protein-ligand binding, one of the cornerstones of CADD.

We test this on two protein complexes. The first test case concerns Flavin Reductase (see *Figure 7a*), which consists of two chains that have flavin mononucleotide ligands (FMN) and one NAD cofactor bound (2BKJ). Martini 2 parameters and mappings from the GROMOS force field were previously published (*Sousa et al., 2021*). Parameters and mappings have been added to the *Vermouth* database. Subsequently, the system could be converted in one step. During a short simulation, the cofactors remain well bound, indicating that no inappropriate parameters or faulty geometries were generated. Next, we created topologies and starting structures for Lysozyme with a benzene molecule bound (1L84), using the Martini 3 force field (*Figure 7b*). The protein and ligand were again converted in one step and then simulated for a short period. As previously, the ligand stays bound, showing that the protocol generates reasonable starting structures and correct parameters.

To fully leverage this new feature, ligand data files are required to be present. Thus, we implemented mappings and parameters from a previously published small molecule database for the Martini 3 force field (*Alessandri et al., 2022*). The set comprises 43 small molecules, which are often part of drugs or drug precursors. All small molecules have corresponding parameters in the CHARMM ligand database, which allows users to directly convert atomistic CHARMM simulations to Martini. Mapping directly from crystal structures as present for example in the PDB or other force fields is

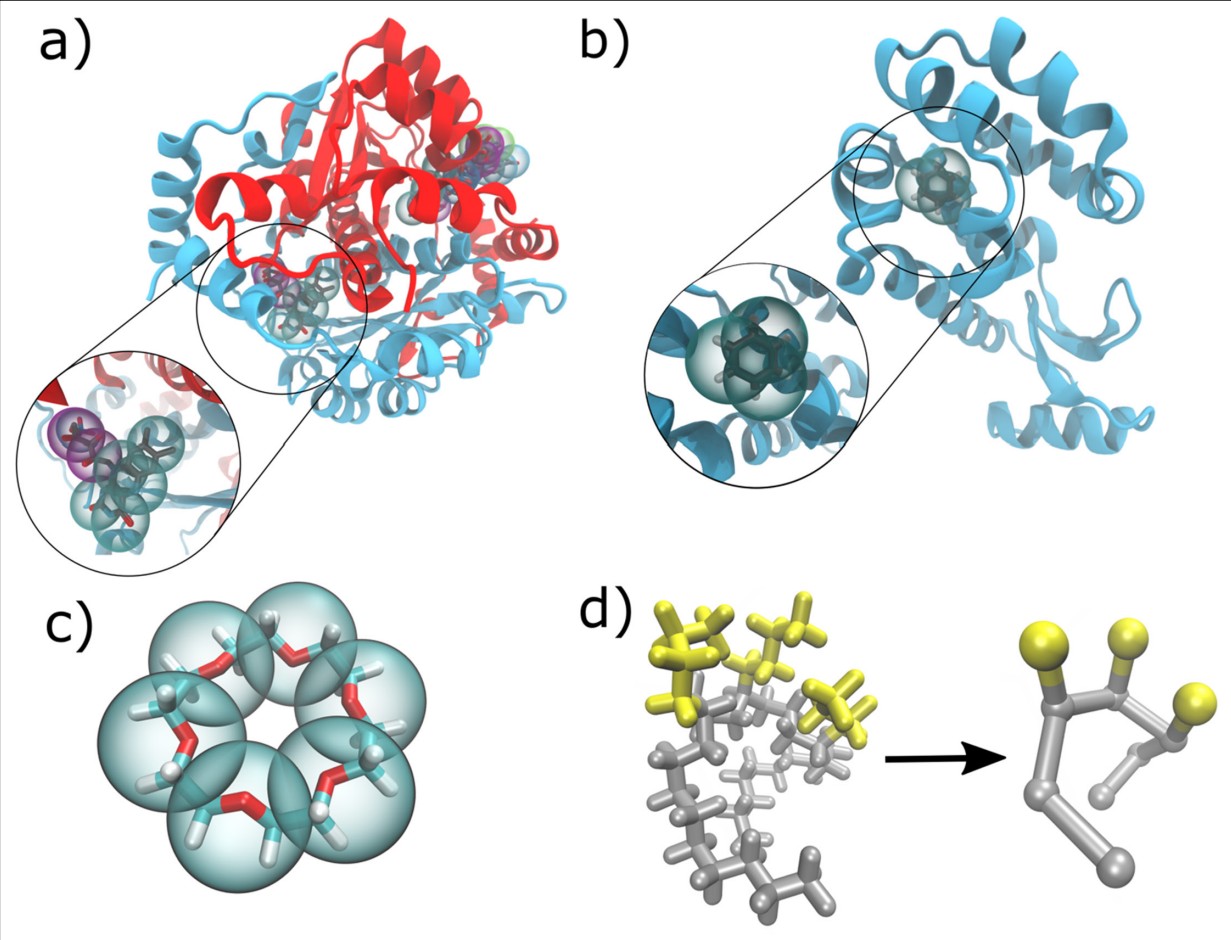

**Figure 7.** Ligands, cofactors, and polymers transformed to CG Martini level. (**a**) Flavin Reductase with two FMNs and one NDP cofactor bound in the reference AA state and mapped to Martini CG as indicated by the spheres. The inset shows a zoom onto FMN; (**b**) Lysozyme with benzene ligand bound in the reference AA structure and mapped to Martini CG resolution; (**c**) Crown ether with Martini beads shown on top of the AA structure; (**d**) Branched polyethylene at AA resolution (left) and Martini resolution (right) with the linear chain part shown in gray and the branches in yellow.

also possible. In these cases, the residue names may have to be adjusted to be the same as in the CHARMM naming convention. However, *Martinize2* is also able to handle topologically more complex molecules. For example, crown ether (*Figure 7c*) consists of six polyethylene glycol (PEO) repeat units and is cyclic. To test whether *Martinize2* can handle cyclic molecules of multiple repeat units, it was converted to Martini 2 resolution applying the latest PEO parameters (*Grunewald et al., 2018*). The second example is branched Polyethylene, where we chose a sequence that begins with two linear units followed by three branched ones and two linear units after. Also, this molecule is converted to Martini 2 resolution *Panizon et al., 2015* by *Martinize2*. Finally, we have set up instructions on how researchers can submit parameters to the database allowing it to grow and support other researchers. In addition, *Martinize2* facilitates dynamic linking of citations to parameters. With this mechanism, citations are printed at the end of the run that dynamically includes citations to all parameters used in the final topology. Such a system also allows researchers to easily receive credit for contributed parameters.

## Complexity benchmark

To assess the robustness of *Martinize2* in a high-throughput use case, we processed the template library used by the I-TASSER (*Yang et al., 2015*) protein prediction software (*Figure 8*). At the time of download (26 March 2021), the dataset contained 87084 protein structures. We processed each of these structures with *Martinize2* to get Martini 2.2 models with ENs. We then energy minimized the

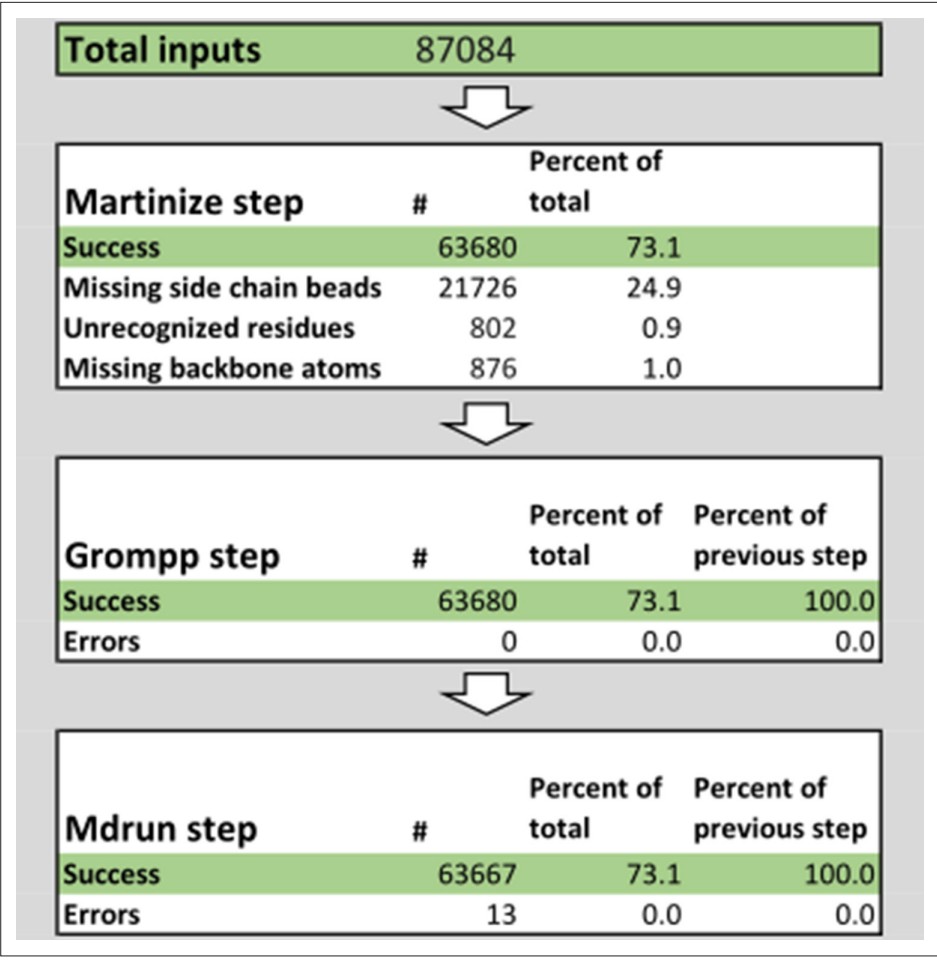

**Figure 8.** Summary of the successes and failures of the high-throughput pipeline. We ran the pipeline on the 87084 structures from the template library used by the I-TASSER (*Yang et al., 2015*) protein prediction software, of which 73% could be converted with Martinize2. The other 26.4% failed mostly due to missing coordinates and unrecognized residues. For 100% of the converted structures, a GROMACS run input file (i.e. tpr-file) could be generated, and on all but 13 of the converted structures, an energy minimization could be performed.

CG protein in a vacuum to validate that the generated structures and topology could be processed by GROMACS 2022.3.

Of the 87,084 structures in the dataset, 63,680 (73%) could be processed through the whole workflow without error. The main cause of failure (25% of the structures) was missing coordinates in the input structures. When all the atoms that compose a bead are missing from the input, *Martinize2* can generate a topology but it cannot generate coordinates for the bead. Note that if only some atoms are missing, then *Vermouth* does estimate the position of the bead. 876 structures (1%) had missing coordinates in the backbone that prevented the use of DSSP (*Kabsch and Sander, 1983*; *Touw et al., 2015*). Finally, 802 input structures (1%) had at least one residue that was inconsistent with the library. Upon further inspection, most of these structures contain malformed glycine residues with an unexpected $C\beta$ atom. *Martinize2* detected these inconsistencies and emitted a warning for each of them; warnings can be explicitly and selectively ignored, and if they are not, no output is written to avoid subsequent workflow steps working with corrupted files.

All the 63,680 input structures that were successfully processed by *Martinize*2 could be processed by the GROMACS pre-processor (*grompp*). However, 13 structures failed the energy minimization. A visual inspection of some of these failing inputs shows the input atomistic structures can be problematic. Erroneous interatomic distances (steric clashes or extended bonds) lead to high energies in the CG systems, which causes a failure in the energy minimization routine. Likely, these starting structures are also not numerically stable in a CG simulation.

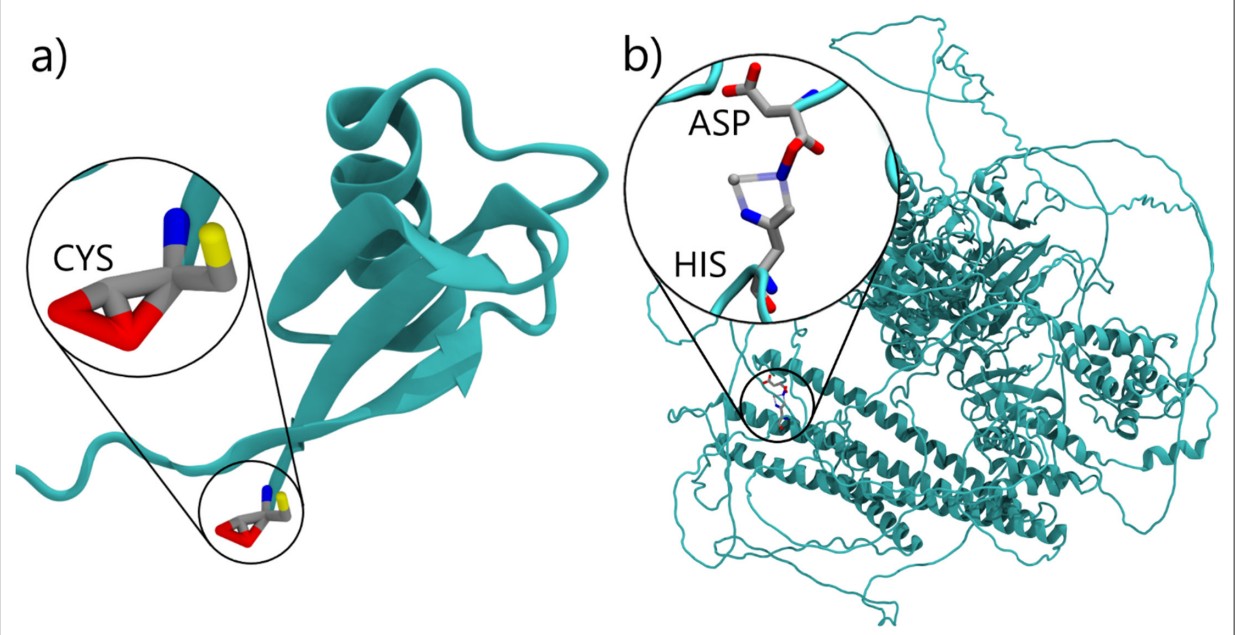

**Figure 9.** Two examples of problematic atomistic protein structures flagged by Martinize2. (**a**) The cysteine residue with too small O-O and O-C distances leads to superfluous bonds being recognized. (**b**) The incorrect interatomic distances in the histidine ring led to missing bonds (transparent), an erroneous O-N bond connecting the histidine to a neighboring asparagine. Additionally, a nitrogen atom is switched for an oxygen atom in asparagine.

As a second test case to assess the robustness of *Martinize2*, we processed a subset of the Alpha-Fold Protein Structure Database (*Varadi et al., 2022*; *Jumper et al., 2021*). 200,000 randomly chosen unique protein structures (see Appendix 5) were given to *Martinize2* and subsequently an energy minimization was performed, if the structure could successfully be converted to CG representation. Of the 200,000 structures in the dataset, 7 structures (see Appendix 5) raised an error during the conversion step. Upon further (visual) inspection of the problematic structures, we concluded that all errors were caused by inaccurate initial atomistic coordinates. These inaccurate atomic positions caused bonds to not be identified or additional superfluous bonds to be detected (*Figure 9*). In these cases, the unrecognizable residues were detected and caused *Martinize2* to emit a warning. The remaining 199,993 successfully converted structures could be processed by the GROMACS pre-processor (grompp), and it was possible to perform an energy minimization.

Finally, as a proof of concept, we tested if the topology generation workflow underlying *Martinize2* is also applicable to generating topologies at the AA level. We selected 100 structures from the AlphaFold database and generated topologies for the CHARMM36 force field. To facilitate the process, a new reader for the .rtp file format, which is the GROMACS protein topology file format, was implemented. In addition, we manually added modifications files to pick up end-terminal modifications, protonation states, and histidine tautomers. *Martinize2* was able to successfully generate topologies for all proteins. Their accuracy was verified by comparing single point energies against topology files generated using *pdb2gmx*. All data and commands for this test case are available from the GitHub examples repository.

## Discussion

In the previous section, we have presented the *Vermouth* python library for facilitating topology generation and manipulation. For researchers to use *Vermouth* as a framework for software development, it presents a clear API separated into data structures, parsers, and processors. In addition, the library relies on only three permissibly licensed open-software projects, namely numpy (*Harris et al., 2020*), scipy (*Virtanen et al., 2020*), and networkx (*Hagberg et al., 2008*). This allows researchers more freedom in licensing their code and reduces the potential for bugs introduced by dependency

changes. Furthermore, the central data structure represents molecules as graphs. Representing molecules as graphs allows us to leverage algorithms from graph theory. Using graph theory for many of the workflows underlying the *Processors* makes them faster and more robust towards edge cases. Even though applying graph theory to molecules is not a new idea (*Engler et al., 2018*; *Engler et al., 2017*; *Cao et al., 2008*), *Vermouth* is specifically designed to also handle CG level molecule transformation focusing on the Martini force field. Therefore, *Vermouth* presents additional functionality often lacking from other packages. For example, the handling of virtual sites, which are ubiquitous in many Martini 3 molecules, is rigorously handled in all *Processors*. As another more general example, the *Processor* applying interactions between residues can automatically compute structural biases from the mapped coordinates. Finally, the *Vermouth* library adheres to the FAIR principles (*Pedersen et al., 2024*; *Chue Hong et al., 2022*; *Wilkinson et al., 2016*) to allow adoption by non-experts and ensure quality control. In particular, for both the *Vermouth* library and *Martinize2*, continuous integration testing is implemented and code review is required. The software is also semantically versioned, and it is distributed through established channels, most notably the Python Package Index, and hosted openly on GitHub.

We have shown how *Vermouth* was used to shape the *Martinize2* program. However, *Martinize2* is not the only program leveraging the power of the *Vermouth* library. The *polyply* Python suite is another library and collection of command line programs built upon *Vermouth*. *Polyply* enables users to generate both AA and CG simulation input data, that is structures and topologies, from sequence information. As such, it allows building system coordinates for arbitrarily complex macromolecular systems and nanomaterials (*Grünewald et al., 2022*). Furthermore, the *martini-sour* package (*Grünewald et al., 2020*), which is currently under development, utilizes *Vermouth* to convert topology files from regular Martini to titratable Martini simulations. *MartiniGlass* uses *Vermouth* to prepare CG topologies for visualization, to further aid evaluation and validation of simulation input topologies (*Brasnett and Marrink, 2025*). These examples already illustrate that *Vermouth* has the potential to indeed become the central framework for Martini software development and possibly for other scientific software developments.

*Martinize2* enables researchers to prepare simulation input files for arbitrary (bio)polymers, starting from an atomistic structure. We have shown in-depth examples focusing on protein-specific applications, given that they are the most important target for *Martinize2*. However, more complex molecules such as cyclic crown ether and branched PE were showcased to demonstrate the capabilities of *Martinize2*. Furthermore, the user has complete control over the data files used. The abstraction of force field data into Blocks, Modifications, and Links allows researchers to reason about model intricacies in a structured manner. This helps the development of optimized models and parameters for complex (polymeric) molecules, as well as clearly defining in which combinations these are validated. The new program uses algorithms from graph theory to identify atoms and assign the appropriate interactions. This makes the program more tolerant towards its input so that the users have to worry less over details such as atom names, or ensuring that all residues are in order and appropriately numbered. In particular, *Martinize2* is capable of detecting and using protonation states, PTMs, and capping groups automatically. In addition, *Martinize2* allows the fine-tuning of the EN and—as it is not limited to proteins—can also generate parameters for ligands, cofactors, or lipids.

In practice, there are decisions a user needs to make when using *Vermouth* and *Martinize2*, especially for high-throughput pipelines. *Martinize2* detects but does not reconstruct atoms that are missing from the input structures; these missing atoms can have adverse effects on the result. In the most harmless cases, they only shift the position of a particle in the output structure. When all the atoms for a particle are missing, then the program cannot compute a position for that particle, leading to an incomplete output where a particle does not have coordinates. Also, some workflows depend on DSSP *Kabsch and Sander, 1983*; *Touw et al., 2015* to assign secondary structures and some specific missing atoms can prevent DSSP from working properly. In those cases, *Martinize2* issues a warning whenever it cannot automatically take care of pitfalls. Handling of these cases is a central difference between the new and old version. The old version either terminates with an undefined error or, probably worse, runs and gives output that does not correspond to the atomistic structure given as input. To illustrate the robustness of *Martinize2* towards problematic input, we applied the program to the complete I-TASSER database (~87 k structures) as well as a subset of the AlphaFold Protein Structure Database (~200 k structures). For the two benchmark cases, *Martinize2* was able to issue a

warning or error for all structures which contained seriously malformed residues. Of the first database, only 13 structures failed in the energy minimization due to problematic starting coordinates but not obviously malformed residues. In the second benchmark set, only seven seriously malformed residues were identified, and all other structures were successfully energy minimized. Thus, we consider *Martinize2* more robust and fit for high-throughput and high-complexity tasks. In addition, *Martinize2* was able to successfully generate CHARMM36 topologies for proteins selected from the AlphaFold database. This proof of concept shows that the workflow underlying *Martinize2* can handle force fields other than Martini.

Ultimately, the robustness comes at a price. *Martinize2* uses subgraph isomorphism to identify atoms based on their connectivity, and then issues a warning or repairs the input. However, subgraph isomorphism is an NP-complete problem (*Cook, 1971*). As a result, *Martinize2* is significantly slower than *martinize*. Nevertheless, considering the flexibility the new program offers, in addition to the fact that it is still fast enough to process all entries in the I-TASSER data bank (*Yang et al., 2015*), this is deemed to be acceptable. Even though *Martinize2* will most likely never be as fast as *martinize,* we note that many of the processes can still be optimized to yield further performance increases. Currently, *Martinize2* is about one order of magnitude slower than its predecessor (*Appendix 3—figure 1*). However, even for large proteins, it takes less than 2 min to generate the input file, which is still much faster than any MD simulation even at the CG level. Aside from the performance limitations, *Vermouth* and *Martinize2* present some other limitations as well. Both are currently only capable of writing topologies in GROMACS format. However, our library does not use the MD parameters of the produced topologies or call GROMACS functions, so support for other MD engines can be added in the future. In addition, since *Vermouth* defines an API, it could even be integrated with existing software such as OpenMM (*Eastman et al., 2017*). Furthermore, the processor pipeline underlying *Martinize2* is currently hardcoded. Future improvements will focus on making the workflow defined by *Martinize2* more flexible, in order to include the processor pipeline as part of the force field definitions. This would enable the use of different pipelines for different force fields, allowing for easier force field-specific post-processing. In addition, implementing CGsmiles (*Grünewald et al., 2025*) as a unified representation of mappings will help offer a broader scope of molecules and make the pipeline more robust with respect to naming conventions in atomistic force fields.

## Methods
### Preparation of protein input files
Crystal structures were obtained from the RCSB for the following proteins (3LZT; 2GS2; 2BKJ; 1L84; 3I40; 3IGM, 1MJ5) or the Alpha Fold Data Bank (*Varadi et al., 2022*) for FtsZ with the ID A0A7Y6D765. Hydrogens and missing heavy atoms were reconstructed using the PRAS package, if appropriate (*Nnyigide et al., 2022*). For 3LST and 1MJ5, the pKa and half-way titration point were estimated using the propka package (*Olsson et al., 2011*). For 3LST, the GLU35 was protonated using the CHARMM-GUI solution builder (*Jo et al., 2008*; *Lee et al., 2016*). The HIS-tag of 1MJ5 was removed.

### All-atom simulations
For 2GS2 and 1L84 CHARMM parameters (*Huang et al., 2017*) were created using the CHARMM-GUI solution builder (*Jo et al., 2008*; *Lee et al., 2016*) and a small equilibration simulation (20 ns) was run before the structures were converted with *Martinize2*. The AA simulation used the recommended non-bonded force settings as for CHARMM with GROMACS (*Bjelkmar et al., 2010*). The temperature was maintained using the v-rescale thermostat by *Bussi et al., 2007* at 310 K and pressure was maintained at 1 bar using the Parrinello-Rahman (*Parrinello and Rahman, 1981*) barostat ($\tau$ =12 ps) after initial equilibration with the *Berendsen et al., 1984* barostat. For the CHARMM36 test case, we subsampled 100 structures from the AlphaFold database and generated CHARMM36 reference itp files using *pdb2gmx*. Subsequently, *Martinize2* was used to generate the same itp file from the coordinates output by *pdb2gmx* to ensure the coordinates are exactly the same. Both itp files were compared by computing a single point energy on the common structure.

## Coarse-grained simulations

All CG MD simulations were run using GROMACS 2021.5 (*Abraham et al., 2015*) and the recommended mdp parameters for Martini 2 (*de Jong et al., 2016*) and Martini 3 (*Souza et al., 2021a*) respectively. In particular, the Lennard-Jones interactions were cut off at 1.1 nm and electrostatics were treated with reaction field (cut-off 1.1 nm, dielectric constant 15). The time step was 20 fs in all cases, and the production trajectories were run with the standard leap-frog integrator. Temperature was maintained using the v-rescale thermostat by *Bussi et al., 2007* at 310 K with ($\tau$ =6 ps) and separate coupling groups for solvent and proteins. The pressure was maintained at 1 bar using the Berendsen barostat for equilibrations ($\tau$ =6 ps). The initial systems were solvated using the *polyply* (*Grünewald et al., 2022*) package or *gmx solvate* utility.

## Complexity benchmark

The (Swiss-Prot) subset of the AlphaFold protein structure database used for the complexity benchmark contained 542,378 pdb structure files at the time of download (22 December 2022). The testing pipeline we used was written in Python and randomly picked 200,000 structures which were given to *Martinize2*. Possible errors during conversion or the subsequent *grompp* and energy minimization steps were captured.

## Acknowledgements

We thank all users that tested the development versions and provided valuable feedback, in particular the members of the SJM group and the participants of the Martini Workshop 2021. We also thank Melanie König for her feedback on the manuscript and figures. Work is supported by an ERC Advanced Grant ("COMP-MICR-CROW-MEM") to SJM. PCTS acknowledges the support of the French National Center for Scientific Research (CNRS) and the research collaboration with PharmCADD. PCTS also thank the PSMN (Pôle Scientifique de Modélisation Numérique) and the Centre Blaise Pascal's IT test platform at ENS de Lyon (Lyon, France) for access to their computing facilities. The platform operates the SIDUS solution developed by Emmanuel Quemener (*Quemener and Corvellec, 2013*). JB acknowledges financial support from the Agencia Estatal de Investigación (Spain), the Xunta de Galicia - Consellería de Cultura, Educación e Universidade (Centro de investigación de Galicia accreditation 2019–2022 ED431G-2019/04 and Reference Competitive Group accreditation 2021–2024, CÓDIGO AXUDA). The European Union (European Regional Development Fund – ERDF) and the European Research Council through consolidator grant NANOVR 866559. CB and SJM acknowledge funding from Novo Nordisk Foundation grant NNF20OC0063808, 'BOUNDLESS'.

## Additional information

### Funding

| Funder | Grant reference number | Author |
|---|---|---|
| European Research Council | COMP-MICR-CROW-MEM | Siewert J Marrink |
| Novo Nordisk | BOUNDLESS | Chris Brasnett |
| Klaus Tschira Foundation | Independent Postdoc | Fabian Grünewald |

The funders had no role in study design, data collection and interpretation, or the decision to submit the work for publication.

### Author contributions

Peter C Kroon, Conceptualization, Methodology, Writing – original draft, Writing – review and editing, software design and implementation; Fabian Grünewald, Supervision, Validation, Methodology, Writing – original draft, Writing – review and editing, software design and implementation; Jonathan Barnoud, Conceptualization, Supervision, Validation, Methodology, Writing – original draft, Writing – review and editing, software design and implementation; Marco van Tilburg, Validation, Writing

– original draft, Writing – review and editing; Chris Brasnett, Supervision, Validation, Methodology, Writing – original draft; Paulo CT Souza, Supervision, Funding acquisition, Validation, Writing – original draft, Writing – review and editing; Tsjerk A Wassenaar, Conceptualization, Supervision, Methodology, Writing – original draft, software design and implementation; Siewert J Marrink, Conceptualization, Supervision, Funding acquisition, Writing – original draft, Project administration, Writing – review and editing

### Author ORCIDs
Fabian Grünewald http://orcid.org/0000-0001-6979-1363
Siewert J Marrink https://orcid.org/0000-0001-8423-5277

Reviewer #1 (Public review): https://doi.org/10.7554/eLife.90627.4.sa1
Reviewer #2 (Public review): https://doi.org/10.7554/eLife.90627.4.sa2
Reviewer #3 (Public review): https://doi.org/10.7554/eLife.90627.4.sa3
Author response https://doi.org/10.7554/eLife.90627.4.sa4

## Additional files

### Supplementary files
MDAR checklist

### Data availability
All code can be found online at https://github.com/marrink-lab/vermouth-martinize (copy archived at *Kroon et al., 2025*). In addition, all released versions are also published on the Python Package Index at https://www.pypi.org/project/vermouth. The documentation is available at https://vermouth-martinize.readthedocs.io/en/latest/index.html. Input files and commands required to reproduce the example test cases from this paper are available on GitHub at https://github.com/marrink-lab/martinize-examples (copy archived at *marrink-lab, 2025*).

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

## Appendix 1

### Input Parsers and Output Writers

**Appendix 1—table 1.** Data Parsers object returned as well as format definition and extension.

| Extension | Data class | Parser name | Input format |
| --- | --- | --- | --- |
| .ff | Links<br>Block<br>Modifications | read_ff | in house force-field format |
| .itp | Block | read_itp | GROMACS topology file; all [molecule] directive content |
| .map | Mapping | read_mapping | mapping file as defined using backwards style |
| .pdb | System | read_pdb | canonical PDB format |
| .gro | Molecule | read_gro | Gromacs.gro file |

**Appendix 1—table 2.** Data Writers and the object returned as well as format definition and extension.

| Input format | Data class | Parser name | Output format |
| --- | --- | --- | --- |
| .gro | System | write_gro | G96 gro file |
| .pdb | System | write_pdb | PDB file |
| .top | System | write_top | Pseudo topology file |
| .itp | System | write_itp | GROMACS topology file; all [molecule] directive content |

## Appendix 2

## Related tools

**Appendix 2—table 1.** Limited overview of selected competing tools capable of generating MD topologies.

'Force Field' lists the force fields for which this tool can generate topologies without changing the source code. 'Type of system' describes the type of system this tool can generate topologies for. 'External data files' means whether the force field parameters used are included in separate data files, making it possible to easily change them. 'Notes' lists additional remarks and comments, 'builds coordinates' means it is capable of constructing coordinates for complete systems, rather than only for e.g. missing side chains.

| Name | Force field | Type of system | External data files | Notes |
|---|---|---|---|---|
| pdb2gmx (*Abraham et al., 2015*; *Páll et al., 2015*) | Any AA/UA | Linear polymers | Yes | |
| LEaP (*Case et al., 2005*) | Any AA/UA | Linear polymers | Yes | |
| CHARMM (*Brooks et al., 2009*) | Any AA/UA | Linear polymers | Yes | |
| Psfgen (*Phillips et al., 2005*) | Any AA/UA | Linear polymers | Yes | |
| Martinize 1 (*de Jong et al., 2013*; *Uusitalo et al., 2015*) | Martini | Proteins, DNA | No | |
| Sirah Tools (*Machado and Pantano, 2016*) | Any CG | Linear polymers | Yes | Performs mapping only |
| DoGlycans (*Danne et al., 2017*) | AMBER, OPLS | Sugars | Yes | Builds coordinates |
| HOOBAS (*Girard et al., 2019*) | Multiple | Multiple | Yes | No user interface, builds coordinates |
| CHARMM-GUI (*Jo et al., 2017*; *Qi et al., 2015*; *Jo et al., 2008*) | Multiple | Multiple | No | Web server, builds coordinates |
| VerMoUTH/Martinize2 | Multiple | Multiple | Yes | This work |
| ATB (*Malde et al., 2011*; *Canzar et al., 2013*) | GROMOS54a7 GROMOS54a8 | Small molecules | N/A | Automatic de novo parametrization |
| LigParGen (*Jorgensen and Tirado-Rives, 2005*; *Dodda et al., 2017b*; *Dodda et al., 2017a*) | OPLS-AA | Small molecules | N/A | Automatic de novo parametrization |
| CGenFF (*Vanommeslaeghe and MacKerell, 2012*) | CHARMM General Force Field | Small molecules | N/A | Automatic de novo parametrization |

## Appendix 3

### Martinize2 pipeline

In this section, we describe the pipeline underlying the martinize2 program in more depth, highlighting the algorithms used.

### Step 1: Parse input

Reading different input file formats is trivial, and all that is needed is to select the correct parser based on the file name provided. At the time of writing, parsers are available for *pdb* and *gro* files (coordinate files in Gromacs format). The input is commonly a list of atoms with associated properties such as atom names, coordinates, and MRU (monomeric repeat unit) names. Sometimes the input also provides information about bonds in the system, such as PDB 'CONECT' records. These will be used if available. Otherwise, bonds will be added between the atoms based on simple geometric criteria. At the very least, we require MRU names and numbers, elements, and either coordinates or bonds. In the end, the input has been parsed and transformed into an undirected graph with atoms as nodes and bonds as edges.

### Step 2: identify and repair

To identify the parsed atoms, the current generation of tools takes the combination of atom name and MRU name as leading, even though this is the most variable between models. For instance, the atom names assigned in the experimental data often do not match the atom names expected by the force field, causing existing tools to either throw an error or even produce incorrect output. We identify atoms based on their MRU names, connectivity, and their elements by overlaying the MRU with its canonical form (*Figure 3* main paper).

Doing so allows us to identify deviations from the canonical structure, such as PTMs, different protonation states, and capping groups. In addition, this method reveals which atoms are missing in the input data, allowing us to reconstruct them. We rely on graph theory to perform the overlaying of input and reference structures (see the dedicated section on graph algorithms below).

In order to do this, every MRU in the input molecule is overlaid with its canonical reference structure with the constraint that the elements of corresponding atoms must be the same. To get the relevant canonical structure, it is assumed the MRU names in the input molecule are correct and that for each MRU, a corresponding block can be found in the library. If the corresponding block cannot be found, an error is raised, and execution is terminated. Since the library files are designed to be human-readable and writable, users can add any data to the library they need.

In the best case, finding the overlay is an induced subgraph isomorphism problem where $M_r \subsetneq R_r$ with $M_r$ an MRU of the input molecule and $R_r$ the corresponding canonical form. However, this is treated as a largest common induced subgraph problem (see below) since $M_r$ can contain 'unexpected' atoms not described by $R_r$ such as PTMs or capping groups. If there are multiple solutions, the solution where most atom names correspond is taken. Either way, a correspondence between the input molecule and its canonical form is obtained. This correspondence is used to (a) identify and add missing atoms, (b) correct the atom names for the atoms that are there, and (c) find which atoms are not described by the canonical MRUs. It should be noted that in this paradigm PTMs, non-standard protonation states, termini, and capping groups are all considered unexpected atoms and treated the same way.

Next, we try to identify all these unexpected atoms by overlaying them with modification template graphs from the library (*Figure 3b* main paper). This is a graph covering problem where we aim to find a minimal combination of templates that covers all unexpected atoms (see below). This does mean that unless there is clear additional metadata, there can be no missing atoms in the found modifications since it is not known what they should look like beforehand. The found correspondences are then used to correct the atom names. The MRUs these atoms are part of are labeled so that the correct mappings and interactions can be applied later on. In the end, the input molecule is complete, has correct atom names, and MRUs that deviate from the reference are labeled. At this point, all information contained in the atom definitions in the input file and their connectivity has been used. Any atoms that could not be recognized will be removed. A warning is issued to the user if this is the case.

## Step 3: resolution transformation

The resolution transformation step maps the input molecule to the desired output resolution (*Figure 3c* main paper). We must assume that these mappings are many-to-many correspondences and that in a mapping from, for example AA to CG, a single AA atom can be mapped to multiple CG beads. Unfortunately, this generalization prevents the use of methods developed in graph theory for this problem so far (*Webb et al., 2019*; *Chakraborty et al., 2018*). Instead, we perform the transformation using the same type of overlay we used to identify atoms in the input molecule. This requires a 'Mapping' object, which consists of two molecular fragments at different resolutions and a correspondence between their particles. These Mapping objects are taken from a library. Including this resolution and transformation step in the pipeline makes vermouth resolution agnostic, capable of also generating CG topologies.

Mappings from the input force field to the required output force field are taken from the library. However, since these mappings can cross MRU boundaries, this is a graph covering problem. This is a variant of the exact cover problem and therefore an NP-hard problem (*Cook, 1971*; *Karp, 1972*). Because in this case it applies to the full polymer, this is intractable. We sidestep this problem by approaching it as if it were an induced subgraph isomorphism problem where all possible places a mapping fits on the input graph are found, respecting the constraints that atom and MRU names must match. In addition, the mapping may only cross MRU boundaries where it is explicitly allowed by the mapping. If mappings overlap, an error is raised. For every mapping that is applied, interactions from the corresponding Block are added to the output graph.

Once done, the found modifications can be mapped. First, the modifications are grouped together by connectivity with their MRUs. This is done because with multiple modifications for a single MRU, their interactions may influence each other, for example (partial) charges in zwitterionic amino acids. Based on these groups, the modification mappings that apply to most of those modifications at once are found by solving the exact set covering problem. The found modifications are then applied by finding the corresponding subgraph isomorphisms. Warnings are issued if multiple modification mappings affect the same particle or interaction.

## Step 4: create topology

Left then is generating the topology. Generating the inter-MRU interactions by applying the appropriate Links is a series of induced subgraph isomorphism problems where all possible ways a link fits on the produced output graph are found. A link can be applied multiple times and can overlap with other links. Whenever a link is applied, the interactions it defines are added to the output graph. In addition to adding interactions, links can also change interactions already set by blocks. For example, the particle type or partial charge may depend on neighboring MRUs. Because of this, links are non-commutative, and the order in which they are applied matters. To resolve this, we solve the subgraph isomorphism problems in the order the links are defined in the library (*Figure 3d* main paper).

At this point, the output graph represents a molecule at the desired resolution with most interactions defined and coordinates that can be generated. Usually, these can be trivially taken from the input coordinates. However, in case atoms were missing in the input, this might not be possible. In those cases, we generate coordinates based on the coordinates of the neighboring atoms.

## Step 5: post-processing

Post-processing can consist of any number of steps and can perform all sorts of force field specific dress-up. For example, it can create an elastic network (*Periole et al., 2009*), or generate the parameters required for Gō interactions (*Poma et al., 2017*). These steps have access to the complete molecule with coordinates and canonical atom names, even if they were missing in the input, and they have access to the full topology with all associated interactions. Separating these steps out into separate Processors helps to keep them independent of each other, and allowing for any type of post-processing helps in making the program force field agnostic. There can be any number of this kind of processors depending on what was requested by the user.

## Step 6: write output

Lastly, the output topology and coordinate files have to be written. Since this is just a matter of file formatting, this is trivial. Separating it out from the rest of the pipeline makes the program agnostic of the MD engine used. At the time of writing, vermouth is capable of writing Gromacs-compatible topologies.

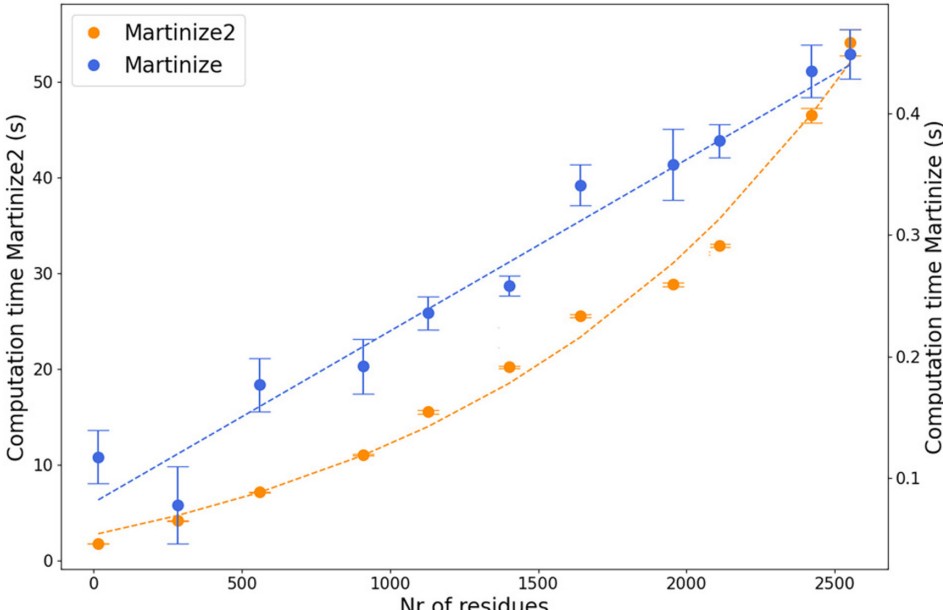

**Appendix 3—figure 1.** Comparison of processing speeds between Martinize and Martinize2. 11 Protein structures with increasing size from a dataset of 200,000 structures were processed with Martinize and Martinize2 to record the computation time. The protein structures were spaced evenly by residue count.

## Appendix 4

### Graph algorithms

Steps 2–4, which form the core of vermouth, rely heavily on graph algorithms, because molecules and polymers can be very naturally described as undirected graphs (*Engler et al., 2018*; *Engler et al., 2017*; *Cao et al., 2008*; *Chung, 2010*). In our case, nodes correspond to atoms, and edges to bonds between atoms. In addition, polymers can also be described as a coarser graph, where nodes correspond to MRUs and edges to bonds between MRUs. Graph theory is a subfield of mathematics that deals with graphs, making it a particularly powerful tool in the context of this work. We primarily use methods from graph theory to identify atoms. First, when curating the provided input data (Step 2), but also when performing the resolution transformation (Step 3) and when applying links (Step 4). Our primary tools for this are algorithms for finding induced subgraph isomorphisms (*Bonnici et al., 2013*; *Houbraken et al., 2014*; *Cordella et al., 2001Demeyer et al., 2013*), and for finding largest common induced subgraphs (*Cordella et al., 2004*; *Krissinel and Henrick, 2004*).

### Largest common induced subgraph

When repairing the provided molecule correspondences between the MRUs in the input molecule ($M_r$) and their canonical forms ($R_r$) are needed. In the case where $M_r$ is not a subgraph of $R_r$ and contains atoms that are not described by $R_r$, this is a largest common induced subgraph (LCIS) problem. The solution to this problem is the largest graph G that is an induced subgraph of both $M_r$ and $R_r$, and the correspondences between the nodes in G and $M_r$; and between the nodes in G and $R_r$. This problem belongs to the class of NP-hard problems (*Cook, 1971*). A possible solution to the LCIS problem is to approach it as a repeating subgraph isomorphism problem where initially $G=M_r$, and nodes are removed from G in a breadth-first manner until an induced subgraph isomorphism $G \subsetneq R_r$ is found (*Krissinel and Henrick, 2004*). Once a subgraph isomorphism between G and $R_r$ is found, the subgraph is not shrunk further since that would always result in a smaller common subgraph. We have based our implementation on the ISMAGS subgraph isomorphism algorithm (*Houbraken et al., 2014*; *Cordella et al., 2001*) since, generally, molecules can be described as very sparse and (locally) symmetric graphs. The ISMAGS algorithm exploits these properties and produces only symmetrically distinct answers, which reduces the runtime significantly compared to both other subgraph isomorphism algorithms, such as VF2 and other LCIS algorithms (*Houbraken et al., 2014*; *Koch, 2001*). Since our implementation of the ISMAGS is more generally applicable than just in the context of vermouth, we have collaborated with the authors of the popular Python graph library NetworkX *Hagberg et al., 2008* to include our implementation.

We extended our implementation of the ISMAGS algorithm to also solve the LCIS problem in order to further exploit the symmetry breaking constraints used in the subgraph isomorphism problem. The symmetry-breaking constraints are used when finding subgraph isomorphisms (see reference 96) and when shrinking the subgraph: when nodes are equivalent, the node with the highest index is removed from G preferentially. In addition, to ensure common subgraphs are preferentially found using nodes with a lower index (analogous to the ISMAGS algorithm), the candidate subgraphs are sorted by their node indices. In this way, we obtain good performance because in our case, it is generally true that: (a) there are only a few nodes not part of the reference, and (b) those nodes have the highest node index. Because of this, we can terminate the algorithm after the first common subgraph is found.

To demonstrate how this works, we consider an example where we will try to find all LCISs between graph X and subgraph Y. The example is illustrated in *Appendix 4—figure 1*. Note that at this point, the distinction between 'graph' and 'subgraph' is arbitrary, except for symmetry detection and performance. Nodes are represented by a letter that reflects their underlying attributes (e.g. atom type). We will consider nodes compatible if they have the same letter, and we distinguish nodes with the same letter by subscripts. First, all symmetries in subgraph Y are found. This reveals $A_1$ to be equivalent to $A_2$. In the first iteration, we try to find a subgraph isomorphism between X and Y (Iteration 1). Since none can be found, subgraph Y is shrunk. This yields the subgraphs in box 'Iteration 2'. Since the subgraph made from the nodes $\{A_1, B, E, F\}$ is symmetry equivalent to the subgraph made from nodes $\{A_2, B, E, F\}$, only the first is taken into consideration. Because no subgraph isomorphism can be found between X and any of these four subgraphs for this iteration,

they are shrunk further, resulting in seven subgraphs with unique symmetries consisting of three nodes each. These are depicted in box 'Iteration 3'. Of these seven subgraphs, at least one is isomorphic to a subgraph of X ({$A_1$, $A_2$, B}), therefore all subgraph isomorphisms between X and these seven subgraphs are exported in order and the algorithm is terminated.

The algorithm presented is not without faults, however: symmetry of X is not taken into consideration, which could reduce the search space dramatically depending on the graphs in question. In addition, some operations are performed multiple times. As an example, many of the subgraphs in *Appendix 4—figure 1* contain the motif {$A_1$, B} (in bold). This results in the subgraph isomorphism algorithm reaching the conclusion that {$A_1$, B} is isomorphic to {$A_1$, B} and {$A_2$, B} multiple times. This can be avoided by starting the algorithm using small subgraphs and growing them as the algorithm progresses. The results of the smaller isomorphism problems can be used to restrict the search space of the larger ones. Since in most of our cases, $M_r$ contains only a few nodes that are not isomorphic to nodes in the $R_r$, we do not expect a (large) performance gain. It may be worthwhile to implement an adaptive algorithm that switches strategy after a few iterations of either strategy, however.

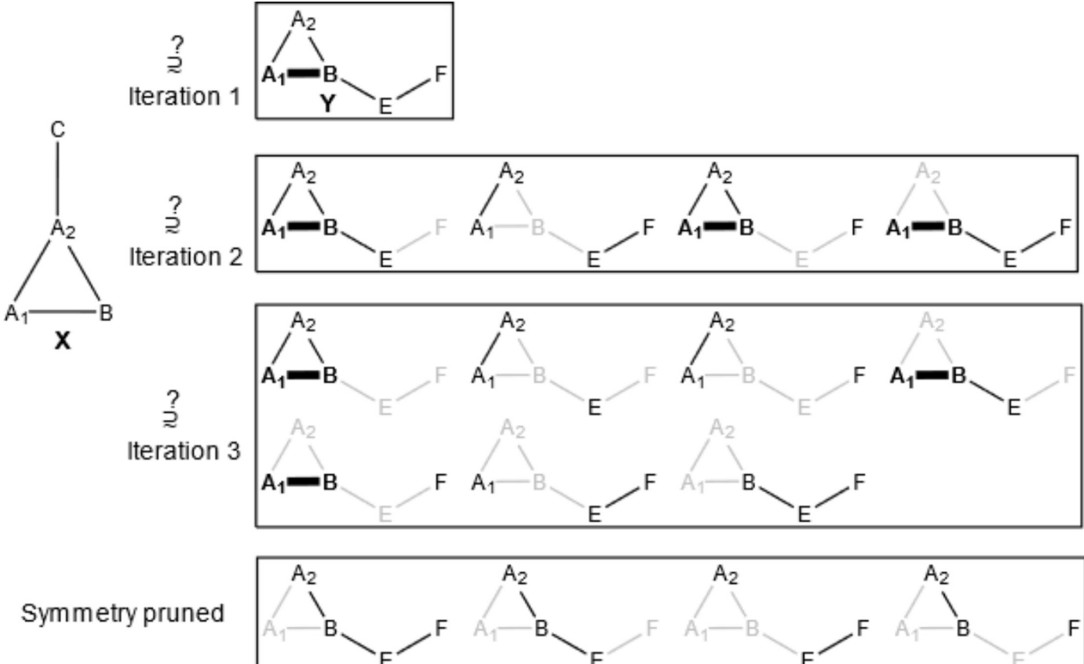

**Appendix 4—figure 1.** Example of finding all LCISs between graphs X and Y. Grayed out nodes are not used (they are excluded from the comparison by the shrinking step), but are depicted for clarity. Since nodes $A_1$ and $A_2$ in Y are symmetry equivalent, not all subgraphs are taken into account. Those that are excluded due to symmetry reasons are depicted in the box Symmetry pruned. Iteration 1: We try to find a subgraph isomorphism between X and Y. None is found. Iteration 2: Y is shrunk to produce the graphs depicted. We try to find subgraph isomorphisms between these and X. None are found. Iteration 3: all graphs from iteration 2 are shrunk further. Since a subgraph isomorphism can be found between at least one of these ({$A_1$, $A_2$, B}) and X, the algorithm terminates afterwards. To highlight how often the algorithm discovers that {$A_1$, B} is subgraph isomorphic to X, it is shown in bold.

## Graph covering

To identify unexpected atoms, we need to cover all those atoms with known fragments describing, for example PTMs. We aim to find the solution where all unexpected atoms are covered exactly once, preferentially using fragments with a lower index. In vermouth, we sort the fragments by size so that larger fragments are used preferentially. This is a variant of the exact cover problem, making it NP hard (*Cook, 1971*; *Karp, 1972*). We solve this problem by a recursive backtracking algorithm: in order, we try to fit the fragments on the unexpected atoms until all are covered. If applying a fragment results in atoms that can no longer be covered, the solution is rejected, and the next fit is tried.

## Appendix 5

### AlphaFold benchmark

The following 7 structures from the AlphaFold benchmarks produced an error, which led martinize2 to abort the input file generation:

AF-O80995-F1-model_v3.pdb
AF-Q58295-F1-model_v3.pdb
AF-B1GZ76-F1-model_v3.pdb
AF-A1ZA47-F1-model_v3.pdb
AF-J9VQ06-F1-model_v3.pdb
AF-F1QWK4-F1-model_v3.pdb
AF-P64653-F1-model_v3.pdb
A list of all surveyed models is available here.

