## [Editor Report · eLife Assessment]

The authors present an **important** multi-scale computational platform, which aims to automate the workflow for coarse-grained simulations of biomolecules in the framework of the popular MARTINI model. The capability of the platform has been **convincingly** demonstrated by the application to a large number of proteins as well as macrocycles and polymers. This work will be of interest to both computational biophysicists and chemists.

---

## [Referee Report · Reviewer #1 (Public review)]

Summary:

In this study, the authors provide a new computational platform called Vermouth to automate topology generation, a crucial step that any biomolecular simulation starts with. Given a wide arrange of chemical structures that need to be simulated, varying qualities of structural models as inputs obtained from various sources, and diverse force fields and molecular dynamics engines employed for simulations, automation of this fundamental step is challenging, especially for complex systems and in case that there is a need to conduct high-throughput simulations in the application of computer-aided drug design (CADD). To overcome this challenge, the authors develop a programing library composed of components that carry out various types of fundamental functionalities that are commonly encountered in topological generation. These components are intended to be general for any type of molecules and not to depend on any specific force field and MD engines. To demonstrate the applicability of this library, the authors employ those components to reassemble a pipeline called Martinize2 used in topology generation for simulations with a widely used coarse-grained model (CG) MARTINI. This pipeline can fully recapitulate the functionality of its original version Martinize but exhibit greatly enhanced generality, as confirmed by the ability of the pipeline to faithfully generate topologies for two high-complexity benchmarking sets of proteins.

Strengths:

The main strength of this work is the use of concepts and algorithms associated with induced subgraph in graph theory to automate several key but non-trivial steps of topology generation such as the identification of monomer residue units (MRU), the repair of input structures with missing atoms, the mapping of topologies between different resolutions, and the generation of parameters needed for describing interactions between MRUs. In addition, the documentation website provided by the authors is very informative, allowing users to get quickly started with Vermouth.

Weaknesses:

Although the Vermouth library can work for different force fields, exhibiting certain generality, its application has been demonstrated only with GROMACS. The extension of the library to other major MD engines could be future directions for improvement but may not be needed for this study.

---

## [Referee Report · Reviewer #2 (Public review)]

This work introduces a Vermouth library framework to enhance software development within the Martini community. Specifically, it presents a Vermouth-powered program, Martinize2, for generating coarse-grained structures and topologies from atomistic structures. In addition to introducing the Vermouth library and the Martinize2 program, this paper illustrates how Martinize2 identifies atoms, maps them to the Martini model, generates topology files, and identifies protonation states or post-translational modifications. Compared with the prior version, the authors provide a new figure to show that Martinize2 can be applied to various molecules, such as proteins, cofactors, and lipids. To demonstrate the general application, Martinize2 was used for converting 73% of 87,084 protein structures from the template library, with failed cases primarily blamed on missing coordinates.

I appreciate the changes that the authors made to clarify the novelty. I have no doubt this paper will receive attention and citations.

---

## [Referee Report · Reviewer #3 (Public review)]

Summary:

The manuscript Kroon et al. described two algorithms, which when combined achieve high throughput automation of "martinizing" protein structures with selected protonation states and post-translational modifications.

The authors have addressed all of my concerns as provided previously. Specifically, Figure S2 will be a very useful guideline for future improvement (e.g., parallelization) of the code.

---

## [Author Response]

The following is the authors’ response to the previous reviews.

**Reviewer #1 (Public Review):**
Summary:In this study, the authors provide a new computational platform called Vermouth to automate topology generation, a crucial step that any biomolecular simulation starts with. Given a wide arrange of chemical structures that need to be simulated, varying qualities of structural models as inputs obtained from various sources, and diverse force fields and molecular dynamics engines employed for simulations, automation of this fundamental step is challenging, especially for complex systems and in case that there is a need to conduct high-throughput simulations in the application of computer-aided drug design (CADD). To overcome this challenge, the authors develop a programing library composed of components that carry out various types of fundamental functionalities that are commonly encountered in topological generation. These components are intended to be general for any type of molecules and not to depend on any specific force field and MD engines. To demonstrate the applicability of this library, the authors employ those components to re-assemble a pipeline called Martinize2 used in topology generation for simulations with a widely used coarse-grained model (CG) MARTINI. This pipeline can fully recapitulate the functionality of its original version Martinize but exhibit greatly enhanced generality, as confirmed by the ability of the pipeline to faithfully generate topologies for two high-complexity benchmarking sets of proteins.Strengths:The main strength of this work is the use of concepts and algorithms associated with induced subgraph in graph theory to automate several key but non-trivial steps of topology generation such as the identification of monomer residue units (MRU), the repair of input structures with missing atoms, the mapping of topologies between different resolutions, and the generation of parameters needed for describing interactions between MRUs. In addition, the documentation website provided by the authors is very informative, allowing users to get quickly started with Vermouth.Weaknesses:Although the Vermouth library is designed as a general tool for topology generation for molecular simulations, only its applications with MARTINI have been demonstrated in the current study. Thus, the claimed generality of Vermouth remains to be exmained. The authors may consider to point out this in their manuscript.

In order to demonstrate generality of the here proposed concepts for generating topologies for molecular dynamics simulations, we have now implemented and tested a workflow that will produce topologies for the popular CHARMM36 all-atom force field. To facilitate generation of all-atom topologies with Martinize2 a .rtp reader was introduced, which allows users to provide .rtp files that are the native GROMACS topology files for proteins instead of .ff files. These .rtp files exist for all major atomic protein forcefields. In addition, for CHARMM36 we also included modification files, which describe non-standard pH amino acids, histidine tautomers, and end terminal modifications. Thus, the current implementation unlocks all features available at the CG Martini level also for CHARMM36. We note that users must add the modifications files for other all-atom force fields e.g. AMBER.

We have added a new item in the main manuscript (p28) briefly describing this proof-of-concept implementation. However, we like to point out that there are many specialized tools for the various force fields adopted by the respective communities. Thus, an exhaustive discussion on the capabilities of Martinize2 for all-atom force fields seemed out of place.

**Reviewer #2 (Public Review):**
This work introduces a Vermouth library framework to enhance software development within the Martini community. Specifically, it presents a Vermouth-powered program, Martinize2, for generating coarse-grained structures and topologies from atomistic structures. In addition to introducing the Vermouth library and the Martinize2 program, this paper illustrates how Martinize2 identifies atoms, maps them to the Martini model, generates topology files, and identifies protonation states or post-translational modifications. Compared with the prior version, the authors provide a new figure to show that Martinize2 can be applied to various molecules, such as proteins, cofactors, and lipids. To demonstrate the general application, Martinize2 was used for converting 73% of 87,084 protein structures from the template library, with failed cases primarily blamed on missing coordinates.I was hoping to see some fundamental changes in the resubmitted version. To my disappointment, the manuscript remains largely unchanged (even the typo I pointed out previously was not fixed). I do not doubt that Martinize2 and Vermouth are useful to the Martini community, and this paper will have some impact. The manuscript is very technical and limited to the Martini community. The scientific insight for the general coarse-grained modeling community is unclear. The goal of the work is ambitious (such as high-throughput simulations and whole-cell modeling), but the results show just a validation of Martinize2. This version does not reverse my previous impression that it is incremental. As I pointed out in my previous review (and no response from the authors), all the issues associated with the Martini model are still there, e.g. the need for ENM. In this shape, I feel this manuscript is suitable for a specialized journal in computational biophysics or stays as part of the GitHub repository.

We apologize for not fixing the typo; it was fixed but unfortunately got reintroduced in the final resubmitted version. We politely disagree that the goal of the work itself is high-throughput simulations and whole-cell modeling, but the Martinize2 tool is certainly an important element in our ambitions to achieve this. Given the broad interest in these goals by the modeling community in general, we believe this work has a much wider impact beyond the (already large) group of Martini users. Addressing limitations of the Martini model itself, which are certainly there, is clearly not the scope of the current work.

**Reviewer #3 (Public Review):**
The manuscript Kroon et al. described two algorithms, which when combined achieve high throughput automation of "martinizing" protein structures with selected protonation states and post-translational modifications. After the revisions provided by the authors, I recommend minor revision.The authors have addressed most of my concerns provided previously. Specifically, showcasing the capability of coarse-graining other types of molecules (Figure 7) is a useful addition, especially for the booming field of therapeutic macrocycles. My only additional concern is that to justify Martinize2 and Vermouth as a "high-throughput" method, the speed of these tools needs to be addressed in some form in the manuscript as a guideline to users.

We have added some benchmark timings in the manuscript SI and pointed to the data in the discussion part, which addresses the timing. Martinize2 is certainly slower than martinize version 1 as we already pointed out in the previous versions. However, even for larger proteins (> 2000 residues) we are able to generate topologies in about 60s. As Martinize2 runs on a single core, it can be massively parallelized. Keeping this in mind the topology file generation is likely to take up only a fraction in a high-throughput pipeline compared to the more costly simulations themselves.